# Targeting the RNA-Binding Protein HuR as Potential Thera-Peutic Approach for Neurological Disorders: Focus on Amyo-Trophic Lateral Sclerosis (ALS), Spinal Muscle Atrophy (SMA) and Multiple Sclerosis

**DOI:** 10.3390/ijms221910394

**Published:** 2021-09-27

**Authors:** Vittoria Borgonetti, Elisabetta Coppi, Nicoletta Galeotti

**Affiliations:** Department NEUROFARBA, Division of Pharmacology and Toxicology, University of Florence, 50139 Firenze, Italy; vittoria.borgonetti@unifi.it (V.B.); nicoletta.galeotti@unifi.it (N.G.)

**Keywords:** RNA-binding proteins (RPBs), ELAV/Hu, HuR, neurodegenerative disease, neuroinflammation, neuroprotection, glia, demyelination

## Abstract

The importance of precise co- and post-transcriptional processing of RNA in the regulation of gene expression has become increasingly clear. RNA-binding proteins (RBPs) are a class of proteins that bind single- or double-chain RNA, with different affinities and selectivity, thus regulating the various functions of RNA and the fate of the cells themselves. ELAV (embryonic lethal/abnormal visual system)/Hu proteins represent an important family of RBPs and play a key role in the fate of newly transcribed mRNA. ELAV proteins bind AU-rich element (ARE)-containing transcripts, which are usually present on the mRNA of proteins such as cytokines, growth factors, and other proteins involved in neuronal differentiation and maintenance. In this review, we focused on a member of ELAV/Hu proteins, HuR, and its role in the development of neurodegenerative disorders, with a particular focus on demyelinating diseases.

## 1. Introduction

RNA is the link between DNA and proteins and contains all the information recovered from DNA to regulate cellular functions and promote cell survival. The importance of precise co- and post-transcriptional processing of RNA in the regulation of gene expression has become increasingly clear. These are complex and controlled processes that lead to mRNA maturation, which require the intervention and interaction of various components, including RNA Binding Proteins (RBPs) [1].

RNA-bound proteins are a class of proteins that bind single- or double-chain RNA, with different affinities and selectivity, thus regulating the various functions of RNA and the fate of the cells themselves [2]. RBPs accompany and protect RNA from transcription in pre-mRNA in the nucleus to nuclear export, up to translation into the cytoplasm [3]. RBPs interact with RNA by binding to sequences or structural motifs through a restricted and structurally well-defined series of RNA-binding domains (RBDs) [2]. Each RBP can contain one or more RBDs, which typically bind 2–6 nucleotides. Although the binding to this small number of nucleotides is sufficient for a correct interaction, RNA can contain more copies of the same domain, allowing the recognition of a larger and more complex region, and increasing the specificity and affinity [4]. Through various in vitro and in vivo studies, more than 1000 RBPs with different biological functions have been identified in humans and mice [5]. They are involved in the coordination and stabilization of complexes with proteins, the processing and maturation of mRNA, alternative splicing and splicing, polyadenylation, transport and localization within the cell, stabilization, the silencing of mature mRNA and in its degradation [6,7,8].

Therefore, given the numerous processes in which RBPs are involved, a mutation or alteration of their functions can lead to the development of diseases [9], such as neurodegenerative pathologies [10], cardiovascular disease [11], tumors [12], and immune system dysregulation [13]. Despite the breadth of RBPs discovered, many of them have not been characterized yet and their biological function is unknown, and it is likely that many RBPs are still to be discovered [1,5]. It is therefore important to adequately understand their functioning, which could lead to the discovery and development of novel drug candidates [14]. RBPs have been linked to diseases such as Alzheimer’s and Parkinson’s disease [15], autism spectrum [16], schizophrenia [17], and other central nervous system (CNS) disorders. Specifically, the RBPs expressed in the CNS are mainly involved in alternative splicing, the process that allows the obtainment of different isoforms of a protein, starting from a single gene. Indeed, alternative splicing occurs more frequently in the nervous system, rather than in other tissues, due to the complexity of neuronal activities, neuronal growth and to the normal activity of synapses [18]. RBPs are also important to protect mRNA from premature translation and degradation during transport from the neuronal body to dendrites and axons, allowing the de novo synthesis of proteins at the synapse level and guiding the transport itself [7]. Neurons have their own systems for regulating RNA, and embryonic lethal/abnormal visual system (ELAV)/Hu proteins represent one of the most investigated family belonging to the RPBs present in neurons. The ELAV/Hu proteins represent an important family of RBPs and play a key role in the fate of newly transcribed mRNA [19]. In Drosophila, the ELAV gene family encodes RBPs specifically expressed in the nervous system and includes three genes, ELAV, RNA-binding protein 9 (rbp9), and fne (found in neurons). In mammals, the four ELAV-like proteins show a high degree of conservation (70–85% identity). Three members, HuB (erlB), HuC (erlC), and HuD (erlD), are specifically expressed in neurons [20]. They are mainly localized in the cytoplasm, but the presence of some specific transporters, such as the transporting Trn1–2, and of some nuclear factors (APRIL and CRM), mediates their translocation in and out the nucleus [21]. The fourth member, HuR/HuA (elrA), is expressed ubiquitously in the organism and is predominantly located in the nucleus. However, because of its shuttling activity, HuR can also be found in the cytoplasm, and its subcellular distribution can change physiologically during cell senescence, stress signals, and tumor growth [22]. The biological function of ELAV RBPs has been identified in their ability to post-transcriptionally control gene expression through cytoplasmic stabilization and/or the enhancement of the translation of their mRNA targets. Structurally, they consist in two binding sites for RNA and one site for binding polyadenylated sequences, separated by a highly variable region that is involved in the interaction with other proteins and is important for receiving signals on the nucleus-cytoplasmic transport, and vice-versa. This last region is generated by alternative splicing to obtain different isoforms, which, while recognizing the same types of RNA, can interact with each other and with other factors in a different way, which leads to the formation of multimeric complexes of ribonucleoproteins. This heterogeneity is very important, especially in the nervous system.

ELAV proteins bind AU-rich element (ARE)-containing transcripts, which are usually present on the mRNA of proteins such as cytokines, growth factors and other proteins involved in neuronal differentiation and maintenance [23]. The binding of ELAVs to these sequences increases the stability of the RNA because it competes with proteins that promote de-adenylation and, therefore, degradation by exonucleases [24]. In addition to the stabilizing effect, these proteins are also involved in the mechanisms of transport to cellular compartments far from the nucleus, such as axons or dendrites, contributing not only to the differentiation of the nervous system, but also to its plasticity [25]. Moreover, they intervene in the mechanisms of splicing and polyadenylation, influencing not only the transcription of other RNAs, but also self-regulation [17]. Further insights into the function of these proteins could help to understand the mechanisms underlying neuronal development and functioning. Given the nature of RBPs, ELAVs influence the fate of many mRNAs, modulating simultaneously many different physiological functions: nELAVs (HuD, HuB, HuC) accelerate neuronal differentiation and axonal outgrowth and modulate neuronal integrity [26], while HuR promotes inflammation and the proper functioning of the immune system [22].

## 2. Physiological Role of HuR outside the Nervous System

HuR can alter the cellular response to various stimuli (proliferative, stress, apoptotic, differentiation, senescence, inflammatory, and immune) through post-transcriptional adjustments of specific mRNAs [22]. ELAV proteins have been thoroughly investigated for their correlation with the insurgence of cancer. The most extensively studied RBP in cancer is HuR. Malignant tumors express a higher amount of HuR than normal tissues or benign tumors [27,28]. Moreover, high levels of HuR are directly related to the development of larger malignant tumors [29] and chemoresistance to standard cancer therapies [30]. HuR interacts with and/or regulates many transcripts involved in the development of cancer cells, such as cyclins, matrix metalloproteinase-9 (MMP-9), metastasis-associated protein 1 (MTA1), anti-apoptotic protein pro-thymosin α (ProTalpha) and HIF-1α [31]. Indeed, its downregulation leads to the reduction of tumor masses [32]. HuR can modulate the expression of proteins involved in cell proliferation, angiogenesis, and myogenesis [32]. Another important role of HuR is related to the inflammatory processes. HuR controls several mRNAs encoding pro-inflammatory cytokines (TNF-α and IL-6), stabilizing them and promoting their expression in different cells of the immune system (i.e., fibroblasts, T-cells, and macrophages) [33], and it is involved in several chronic inflammatory pathologies, such as rheumatoid arthritis [34]. Moreover, HuR stabilizes the mRNAs that encode one major pro-inflammatory mediator, the enzyme cyclooxygenase-2 (COX-2) [35], which is expressed in several types of cancer. Indeed, HuR-silencing has been proposed as an adjuvant therapeutic target in chemotherapy for pancreatic cancer [36]. HuR was found to stabilize and increase nitric oxide synthase 2 (NOS2) expression in carcinomas of the lung and colon [37]. Moreover, interleukin-10 (IL-10), a well-known anti-inflammatory cytokine, can reduce the levels, and repress the action, of HuR. Thus, in a human monocytic cell line, IL-10 destabilized the post-transcriptional mRNA expression of inflammatory cytokines, and this effect depended on the ability of IL-10 to inhibit the expression of HuR [38]. HuR has also been shown to play a role in cardiovascular pathologies [11]. It has been reported to possess a functional role in the progression of pathological cardiac hypertrophy [39], which is linked to an abnormal placental labyrinth vasculature [40]. Finally, it has been reported that HuR induces the stabilization of angiogenic factor-encoding, the vascular endothelial-derived growth factor (VEGF), and MMP-9 mRNAs [41]. Through HuR protein, the alphavirus Sindbi reduces mRNA decay and maintain a highly productive infection [42]. Moreover, HuR may be involved in the reverse transcription reaction of the human immunodeficiency virus 1 (HIV-1) [43]. In muscle fibers from inclusion body myositis (IBM) patients, both the poly(A)-binding protein 1 and HuR were found to aggregate in RNA deposits; these observations were suggested to reflect an impairment in mRNA turnover and translation in IBM [44]. During skeletal myogenesis, HuR coordinates the regulation of muscle differentiation genes [45]. The activation of adenosine monophosphate AMP-activated protein kinase (AMPK) results in HuR nuclear sequestration, the inhibition of NOS2 synthesis, and a reduction in cytokines-induced MyoD loss. These results define NOS2 and HuR as critical players in cytokines-induced cachexia, establishing them as potential therapeutic targets [46].

## 3. HuR in the Nervous System: A Key Regulator of Neuroinflammation

Astrocytes and microglia are the first elements to respond to damage in the CNS. In fact, these cells change their phenotypes from an anti- to pro-inflammatory state in response to signals coming from the damaged site itself [47]. Chronic inflammation represents an important protagonist in the development and progression of neurodegenerative diseases. Indeed, a number of pro-inflammatory molecules have been detected in the brains of patients with neurodegenerative diseases [48]. The presence of risk factors (genetic and environmental) leads to continuous and dis-regulated microglial activation, promoting a chronically unregulated cell state. This loss of control through microglial activity leads to the deterioration of the brain’s structure and functions [49]. The process by which microglia change their shape, footprint, and physiology is defined as the activation of microglia, during which these cells can switch from a pro-inflammatory to an anti-inflammatory phenotype [50]. The anti-inflammatory phenotype is responsible for damage repair and immunomodulation. In the case of prolonged neuroinflammation, the ability of microglia to assume this progressive inactivation profile is limited, thus causing a state of chronic activation [51]. Activated microglia can become harmful to the nervous system through the secretion of inflammatory cytokines and chemokines [52,53]. There are numerous neurodegenerative diseases characterized by strong microglial activation, such as Parkinson’s disease, Alzheimer’s disease, and multiple sclerosis (MS). Therefore, chronic neuroinflammation is considered a potential target for innovative therapeutic strategies [54]. Activated astrocytes also play a crucial role in neuroinflammation and, through the release of cytokines and chemokines, perform the main role of recruiting and activating other cells of the immune system, such as neutrophils, monocytes, and microglia itself [55]. There is growing evidence for the existence of heterogeneity among reactive astrocytes across different brain regions, but also locally, within the same region, as regards astrocyte proliferation, morphology and gene expression. Inflammatory mediators can drive astrocyte transcriptome profiles towards pro-inflammatory phenotypes [56] that possess a protective effect against microbial infection but can be harmful in neurodegenerative diseases [57]. Among the inflammatory mediators released by microglia and astrocytes, tumor necrosis factor-α (TNF-α), interleukin-1β (IL-1β), interleukin-6 (IL-6), and matrix metalloproteases are the most common. These factors further activate other glial cells, facilitate their mobility, and attract immune cells circulating in the damaged area. The result is an amplification of the production of pro-inflammatory factors, the accumulation of cytotoxic substances, the loss of integrity of the blood-brain barrier (BBB), and, finally, cell death [58]. All these cellular events are linked to the ubiquitous transcription factor nuclear factor-κB (NF-κB), which is usually inactivated by inhibitory proteins, such as IκBα, in the cytoplasm. The main activators of this factor are the products of cell damage and cytokines (TNF-α and IL-1β) release. In general, NF-κB is involved in a wide range of biological processes underlying inflammation and the immune response [59]. Neuroinflammatory processes in diseases such as MS are closely related to the activity of the immune system. The involvement of CD4 + T lymphocytes in the pathogenesis of many inflammatory and autoimmune diseases has been demonstrated. These cells, once differentiated, assume different functions and produce specific combinations of pro-inflammatory cytokines. Among the various subtypes, Th17, which produces Interleukin-17 (IL-17), plays a role in the pathogenesis of experimental autoimmune encephalomyelitis (EAE) and MS [60]. It has been shown, for example, to stabilize pro-inflammatory cytokines in CD4 + T lymphocytes, leading to an increased response in inflammatory diseases, such as EAE [61]. Often, studies concerning the mechanisms and elements involved in cellular processes are based on the analysis of the levels of mRNA and their protein derivatives. In many cases, there is no proportionality between these two elements, which is why it is not possible to obtain correct information by considering only transcriptomics studies [62]. Matsye et al. [63] demonstrated the involvement of HuR in the molecular and cellular phenotype of activated microglia. Although it is normally found in the nucleus, in the activated microglia of the spinal cord of mice and humans with amyotrophic lateral sclerosis (ALS), HuR is overexpressed and mainly localized in the cytoplasm. The knockdown of HuR in microglial cells led to an attenuation of the expression of factors in inflammation, chemotaxis and cell migration, which is less related to a decrease in mRNA stability. On the contrary, this has been linked to the suppression of the promoter activity of NF-κB transcription. It has also been shown that the involvement of HuR in the translation processes, in some cases, may be independent of the stabilizing effect on mRNA. An example is that the reduction of TNF-α levels is not associated with the reduction of mRNA levels. Similar results were obtained by Kwan et al. [64], who evaluated the role of HuR in activated astrocytes in a model of spinal injury, where the inhibition of HuR leads to a reduction of cytokines and chemoattraction for neutrophiles and microglia [64]. Furthermore, HuR has been evaluated as a specialized controller of oxidative stress in neurons to confer protection from neurodegeneration [65].

## 4. HuR in Neurodegenerative Disorders

In this review, we listed a series of chronic neurodegenerative diseases in which HuR is reportedly involved. The experimental details about each disorder are reported in Table 1.

### 4.1. HuR and Amyotrophic Lateral Sclerosis

Amyotrophic Lateral Sclerosis (ALS) is a neuromuscular disease characterized by motor neuron degeneration, in which the chronic activation of microglia contributes to disease progression [66,67]. Approximately 10% of ALS cases are hereditary, with a mutation of copper-zinc superoxide dismutase 1 (SOD1) being the most frequently identified genetic defect [68]. ELAV proteins are positive regulators of SOD1 gene expression; indeed, the docking sites for these proteins have been found in SOD1 3’-untranslated region [68]. The role of HuR in the pathogenesis of ALS remains controversial. An increase of HuR expression in peripheral blood mononuclear cells (PBMC), together with its upregulation in the motor cortex or the spinal cord in ALS patients has been reported. In parallel, the upregulation of SOD1 has been observed, suggesting that HuR could be involved in the post-transcriptional phase of SOD1 gene expression [69]. Silencing HuR in the microglia of ALS-associated mutant SOD1 mice reduced the release of pro-inflammatory cytokines. These findings highlighted the important role of HuR in the phenotype of activated microglia, which could be exploited as a possible therapeutic target in ALS [63] (see Table 1). Contrary to these results, other evidence suggested that HuR downregulation might lead to an aggravation of disease. Indeed, HuR stabilizes and enhances the translation of growth factors, and of VEGF. It has been reported that HuR increased VEGF expression and reversed the impairment of mitochondrial function and oxidative stress, suggesting that the impairment of HuR may accelerate CNS atrophy and cell death [68]. The role of HuR in regulating two RBPs, namely TAR DNA-binding protein of 43 kDa (TDP-43) and Fused in Sarcoma/Translocated in Liposarcoma (FUS/TLS), has been linked genetically to ALS. About 4% of hereditary cases of ALS are due to mutations in TARDBP, the gene that encodes TDP-43, a nuclear protein that regulates mRNA processing; and glial TDP-43 pathology is present in ALS [70]. Lu et al. [71] demonstrated that HuR regulates TDP-43 expression in astrocytes cells and that this cross-talking influences the phenotype of glia cells. Another important aspect of ALS patients is the presence of severe metabolic dysfunction, characterized by hypermetabolism, hyperlipidemia, and glucose intolerance, which lead to a drastic degeneration of the disease [72]. AMPK plays a key role in controlling metabolism in the CNS, which is indeed modulated by AMP levels and cellular stress. Altered AMPK activation was reported in the spinal cords of mice with mutant SOD1, and abnormal localization of HuR was associated with enhanced AMPK in the motor neurons of ALS patients, inducing an imbalance in the RNA metabolism and contributing to ALS pathogenesis [73]. Energy deficits may contribute to motor neuron vulnerability in ALS, suggesting that the modulation of energy balance may be a potential therapeutic approach.

### 4.2. HuR and Spinal Muscle Atrophy

Spinal Muscular Atrophy (SMA) is a neuromuscular disease characterized by the degeneration of motor neurons in the spinal cord, caused by a mutation in the motor neuron 1 gene, SMN1. This mutation leads to the death of alpha-motoneurons, the reduction of myogenesis and a loss of innervation, causing muscle weakness and atrophy [74]. HuR can stabilize some important transcription factors, such as Myogenin, MyoD [75], acetylcholine receptor beta (AChR β) subunit mRNAs [76], and p21 [45], which are involved in muscle differentiation. To achieve these effects, HuR must accumulate in the cytoplasm. Thus, the critical role of HuR in myogenesis necessarily involves the import factor Transportin 2 (TRN2), which is activated by a caspase-dependent cleavage of a cytoplasmic HuR. Indeed, HuR cleavage generates two fragments: HuR-cleavage product 1 (HuR-CP1) and 2 (HuR-CP2); HuR-CP1 is responsible for HuR accumulation in the cytoplasm [77]. However, HuR-CP1 itself cannot induce myogenesis in the absence of full-length HuR, suggesting that the effect on myogenin mRNA is indirect via HuR, and emphasizing the importance of the remaining uncleaved cytoplasmic HuR in this process as a regulator of gene expression for myogenesis factors. The importance of caspase-3 activation for muscle differentiation has been described. It has been shown that during myogenesis, HuR could be cleaved by this caspase [78]. While the role of HuR in the myogenesis process is partly understood, the role of caspase-3 remains completely unclear. Another pathway that induces the cleavage of HuR involves the stress-response kinase protein kinase dsRNA (PKR). PKR was shown to be important for muscle differentiation because of its regulation of the p38 MAPK and PI3K/Akt pathways [79]. PKR activates the FADD/caspase-8/caspase-3 pathway to trigger HuR cleavage, without being phosphorylated [78]. P38 activators may be valuable candidates for the management of SMA, as inducing p38 phosphorylation leads to an increase of cytoplasmatic HuR levels. In the cytoplasm, HuR can interact with the 3′-UTR of functional survival motor neuron (SMN) transcription [80]. Celecoxib, an FDA-approved p38-activator, increased SMN quantity, improved motor function, and enhanced animal survival, in a mouse model of SMA [81]. P38 is expressed much more in microglia than in neurons and astrocytes. Thus, the effect of HuR could be mediated both by the regulation of myogenesis and by the modulation of microglia activation.

### 4.3. HuR in Multiple Sclerosis

Multiple Sclerosis (MS) is a chronic autoimmune inflammatory and neurodegenerative disease of the CNS, characterized by demyelination, axonal loss and motor dysfunction, that affects 2.9 million people worldwide [82]. Inflammation plays a central role in MS; indeed, pro-inflammatory microglia are associated with the demyelization process [83,84], through the release of cytokines and other mediators of inflammation, as well as oligodendrocyte degeneration and myelin loss [85,86,87]. MS progresses in different stages, beginning with a cascade of inflammation. The pivotal spark to initiate this cascade seems to be the migration of Th17 into the CNS across the BBB through disrupted tight junctions. Consequently, glial cells are activated in the spinal cord, leading to the excessive release of pro-inflammatory cytokines [88]. Therefore, targeting microglia could represent an interesting focus for the development of novel therapeutical approaches. In the EAE model, the repeated intrathecal administration of anti-HuR antisense oligonucleotide (ASO) reduced the main behavioral symptoms in mice, which reproduced the main clinical signs seen in MS patients [89]. The intrathecal administration of anti-HuR ASO decreased the proinflammatory-activated microglia, inflammatory infiltrates, the expression of the proinflammatory cytokines IL-1β, TNF-α, and IL-17, and inhibited the activation of the NF-κB pathway. The beneficial effect of anti-HuR ASO in EAE mice also corresponded to decreased permeability of the BBB. In the spinal cord tissue of EAE mice, HuR expression colocalized with activated microglia, and its inhibition led to a reduction of the microglia pro-inflammatory phenotype [90]. Another important effect was that anti-HuR ASO administration reduced the mechanical allodynia produced by the EAE model. This effect was also observed in a peripheral model of neuropathic pain, in which HuR colocalized with microglia cells and induced a shift from a pro-inflammatory to an anti-inflammatory state [91]. HuR is ubiquitously expressed in PBMC, and its activity has been observed in Th17 cells, which are deeply involved in the encephatoligenic process promoted by MS in the CNS [92]. IL-17-producing Th17 cells are major contributors to chronic inflammatory and autoimmune diseases, such as MS, rheumatoid arthritis, and inflammatory bowel disease. Mice with adoptively transferred HuR KO Th17 cells demonstrated delayed initiation and reduced disease severity at the onset of EAE, compared with those with wild-type Th17 cells [93]. The C-C chemokine receptor 6 (CCR6) is critical for pathogenic Th17 cell migration to the CNS. The silencing of HuR reduced CCR6 expression in Th17 cell and reduced their migration to CNS in the EAE model, compared to the WT group [94]. HuR altered the transcriptome of Th17 cells characterized by reduced levels of differentiation factors (namely, RORγt, IRF4, and T-bet), thereby reducing the formation of pathogenic IL-17+IFN-γ+CD4+ T cells in the spleen. These data have been confirmed by measuring the activity of dihydrotanshinone I (DHTS), a well-known HuR inhibitor, that reduced Th17 differentiation, reducing the symptoms associated with the EAE model [95]. However, in contrast with the above results, PBMCs from 52 MS patients demonstrated a lower HuR protein content compared to 43 healthy controls, which continued to decline with the progression of the disease. These opposite expression levels might indicate that HUR performs different roles depending on whether it is inside or outside the CNS, thus promoting neuroinflammation in the spinal cord and controlling the immune response peripherally [96]. Even though additional clinical data are required, these results highlighted the key role of HuR in the initial pathogenesis of MS, during the initiation of the inflammatory process.

## 5. Pathophysiological Role of nELAV

Growing evidence indicates the prominent role of the neuron-specific members of the ELAV family, the nELAV (HuD, HuC, HuB), in the development of neurodegenerative disorders [97]. Through their mRNA-stabilizing activities, neuron-specific ELAVs (HuB, HuC, HuD) modulate neuronal development, participate in synaptic plasticity mechanisms in the CNS, promote the regeneration of peripheral nerves, and accelerate neuronal differentiation and axonal outgrowth. HuD, the best-studied member of the nELAVs, demonstrated its role in promoting neuroprotection through the regulation of brain-derived neurotrophic factor (BDNF) and GAP43 [89,98]. HuD target mRNAs are associated with several neurological disorders, including neurodegenerative disorders such as Alzheimer’s, Huntington’s, and Parkinson’s disease, mood disorders, epilepsy, schizophrenia, and intellectual disabilities, such as Rett syndrome [26]. Fewer data are available for the physiopathological role of HuB and HuC. HuB has been associated with schizophrenia and autism spectrum disorder. HuC single mutant and haplo-insufficient heterozygous mutant are associated with spontaneous epilepsy. More recent studies correlate HuB and HuC with mood disorders. A subset of HuC binding sites was mapped to the 3′ UTRs of genes involved in pathways regulating glutamate pathway, a major excitatory neurotransmitter in the brain [20]. HuB and HuC silencing promotes neuron generation in the CA3 hippocampal region in a mice model of depression [99]. Other studies showed that mHuB and mHuC induced the ectopic expression of neuronal markers, whereas the dominant-negative forms of mHuB and mHuC suppressed the differentiation of CNS motor neurons. Importantly, evidence indicates that ELAV members act in concert. An interplay was demonstrated between HuR and nELAV RBPs in neuronal differentiation in vitro, through which the appearance of nELAV in neuroblasts downregulated HuR expression [100]. An imbalanced nELAV/HuR protein ratio in the nervous system could on one hand augment inflammatory and immune phenomena, and on the other hand promote axon degeneration. From these observations, all ELAV proteins are involved in neuronal differentiation in the mammalian nervous system [101], and proper neuronal development and maintenance appear to be strictly related to the control of proinflammatory responses. In recent years, the role of neuroinflammation in the onset and promotion of neurological and neurodegenerative disorders has emerged, making ELAVs valuable therapeutic targets.

## 6. Conclusions

Hu proteins represent a heterogeneous family of proteins characterized by different physiological functions. HuR is ubiquitously expressed in different tissues, and for this reason its role has been investigated in different pathologies [102]. HuR can simultaneously modulate the mRNAs involved in inflammatory and regenerative phenomena, which makes it an innovative and promising target. The contribution of HuR in several diseases has been postulated, but its exact role is still poorly understood. At the same time, the variability and the lack of information about the complete pharmacological and physiological profile of Hu proteins limits the development of specific treatments. Glia–neuron interaction is a novel target for innovative therapies. Indeed, microglia can communicate with neurons through the release of several mediators (i.e., growth factors, cytokines) or through direct contact (i.e.,: Notch-1 pathway). Hence, a sort of communication between Hu family members could exist, and could be modified based on the CNS environment. Investigating the robust anti-inflammatory capability of HuR, together with the neuronal effects of nELAV, could represent an important step towards a better understanding of the pathophysiological role of these proteins, and offer a potential target for novel drug candidates.

**Table 1 ijms-22-10394-t001:** In vivo and in vitro investigations on the pathological role of HuR.

Diseases	Models	Tissues/Cells	HuR Inhibitors	HuR Activators	Target	Reference
**Amyotrophic Lateral Sclerosis (ALS)**	ALS-associated mutant SOD1 mice Microglial murine cell murine (BV2)	Microglia	*HuR gene silencing* MS-444 MMP408	-	Inflammation immune function cell migration	[63]
	SOD1-G93A transgenic mice G37R-SOD1 transgenic mice Human glioblastoma astrocytoma (U-251 MG)	Lumbar spinal cord	HuRKO; doxycycline	Doxycycline	VEGF	[68]
	SOD1-G93A transgenic mice Human glioblastoma astrocytoma (U-251 MG)	Cortical primary astrocytes	*HuR gene silencing* MS-444	Doxycycline	TDP-43 FUS/TLS	[101]
	Human spinal cord sections C57BL/6 mice Neuroblastoma hybrid cell line (NSC-34)	Spinal cord tissue	-	-	AMPK	[73]
**Spinal Muscular Atrophy (SMA)**	Mouse Myoblast cell line(C2C12)	-	*HuR gene silencing*	-	myogenesis	[102]
	Human neuron-committed teratocarcinoma (NT2) Mouse Motor Neuron-derived (MN-1)	-	-	Anisomycin	p38	[80]
	Human neuron-committed teratocarcinoma (NT2) Mouse Motor Neuron-derived (MN-1) SMA mice model (SMAΔ7)	Spinal cord	*HuR gene silencing*	Celecoxib	Survival Motor Neuron (SMN) protein	[81]
**Multiple Sclerosis** **(MS)**	EAE: PLP139–151 peptide	Spinal cord	aODN against HuR	-	Clinical signs Demyelization	[84]
	EAE: PLP139–151 peptide	Spinal cord Plasma Brain	aODN against HuR	-	Neuroinflammation	[87]
	HuR KO	CD4+ T cells Mononuclear cells from spinal cords	HuRKO mice	-	IL-17	[90]
	HuR KO	CD4+ T cells Mononuclear cells from spinal cords	HuRKO mice	-	C-Chemokine receptor 6 (CCR6)	[91]
	EAE: MOG35–55 peptide Human naive CD4+ T cell	CD4+ T cells Mononuclear cells from spinal cords	HuRKO mice	-	Th17 Cell Differentiation	[92]
	MS patients	PBMC	-	-	-	[93]

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
