# Peer review of "Targeting the RNA-Binding Protein HuR as Potential Thera-Peutic Approach for Neurological Disorders: Focus on Amyo-Trophic Lateral Sclerosis (ALS), Spinal Muscle Atrophy (SMA) and Multiple Sclerosis"

_ijms, 2021, doi:10.3390/ijms221910394_

Round 1
Reviewer 1 Report
1. The title of the publication, “Targeting the RNA-Binding protein HuR as a novel therapeutic approach for neurodegenerative diseases” does not fully which is written in the article.
In this review the authors only explore three of the neurodegenerative disorders, Amyotrophic Lateral Sclerosis, Spinal Muscle Atrophy, Multiple sclerosis, not even mentioning the HuR involvement with the most common neurodegenerative disorders, such as Alzheimer's or Parkinson's disease.
In my opinion, if they want to keep a general title as it is now they must explore the role of HuR in other neurodegenerative disorders. There is also a lot of new evidences about microglia activation in different Tauopathies and Synucleiopathies, as Alzheimer's and Parkinson's disease.
2. In the chapter 2 the authors mention the importance of different stimulus, as for example cell senescence for HuR and that can alter the cellular response.
Nowadays, more attention is be given to this cell phenomena defined as the cell cycle arrest in which there is a decline of the regenerative potential and function of different tissues.
In the context of neurodegenerative disorders, an increased presence of senescent cells in different disorders is being found, giving more evidences a possible effect of senescence in the pathophysiology of these disorders. There is an extensively body of literature that associates cellular senescence with several neurodegenerative disorders including Alzheimer's disease (AD), Down syndrome (DS), and Parkinson's disease (PD). This review summarizes it https://doi.org/10.3389/fncel.2020.00016.
The authors might explore the subcellular distribution change of HuR during cell senescence and how might be linked with neurodegenerative disorders.
3. In the line 167 the authors state -Inflammatory mediators can drive astrocyte transcriptome profiles towards pro-inflammatory phenotypes [57] that possess a protective effect against microbial infection but can be 169 harmful in neurodegenerative diseases [58].
I suggest to reformulate this sentence. This sentence is to affirmative for which is currently known about the neuroprotective pro-inflammatory phenotype of the astrocytes. There is still a doubt if they have or not a neuroprotective effect. The authors might have a look in this review https://doi.org/10.1186/s40035-020-00221-2.
Author Response
We thank the reviewers for their comments on the manuscript and we hope that the changes made in the manuscript have solidified our conclusions and satisfied the reviewer.
Reviewer n. 1
1.The title of the publication, “Targeting the RNA-Binding protein HuR as a novel therapeutic approach for neurodegenerative diseases” does not fully which is written in the article.
In this review the authors only explore three of the neurodegenerative disorders, Amyotrophic Lateral Sclerosis, Spinal Muscle Atrophy, Multiple sclerosis, not even mentioning the HuR involvement with the most common neurodegenerative disorders, such as Alzheimer's or Parkinson's disease. In my opinion, if they want to keep a general title as it is now they must explore the role of HuR in other neurodegenerative disorders. There is also a lot of new evidences about microglia activation in different Tauopathies and Synucleiopathies, as Alzheimer's and Parkinson's disease.
We thank the review for pointing out this issue. We did not describe the role of HuR in other neurodegenerative diseases because, to the best of our knowledge, its role has not been well understood yet for Alzheimer's and Parkinson's diseases. Indeed the available literature for the Alzheimer’s (doi:10.3233/JAD-2009-0967) and Parkinson’s (doi: 10.1007/s00439-005-1259-2) diseases reports a much more important role for the neuronal Elav protein, in particular the most studied is HuD, an Elav protein expressed on neuronal cells, compared to HuR. However, to accomplish with the reviewer concern, we changed the title and we listed the neurological disorders that we described.
- In the chapter 2 the authors mention the importance of different stimulus, as for example cell senescence for HuR and that can alter the cellular response.
Nowadays, more attention is be given to this cell phenomena defined as the cell cycle arrest in which there is a decline of the regenerative potential and function of different tissues.
In the context of neurodegenerative disorders, an increased presence of senescent cells in different disorders is being found, giving more evidences a possible effect of senescence in the pathophysiology of these disorders. There is an extensively body of literature that associates cellular senescence with several neurodegenerative disorders including Alzheimer's disease (AD), Down syndrome (DS), and Parkinson's disease (PD). This review summarizes it https://doi.org/10.3389/fncel.2020.00016.
The authors might explore the subcellular distribution change of HuR during cell senescence and how might be linked with neurodegenerative disorders.
We thank the reviewer for highlighting this point. We improved the paragraph 3.1 by adding and commenting on the suggested reference.
- In the line 167 the authors state -Inflammatory mediators can drive astrocyte transcriptome profiles towards pro-inflammatory phenotypes [57] that possess a protective effect against microbial infection but can be 169 harmful in neurodegenerative diseases [58].
I suggest to reformulate this sentence. This sentence is to affirmative for which is currently known about the neuroprotective pro-inflammatory phenotype of the astrocytes. There is still a doubt if they have or not a neuroprotective effect. The authors might have a look in this review https://doi.org/10.1186/s40035-020-00221-2.
We thank the reviewer for the suggestion. We reformulated the sentence at lines 197 and 198.

Reviewer 2 Report
The authors reviewed the physiological role of HuR outside and inside the nervous system, and the relationship of HuR with neurodegenerative disorders. They have especially focused on the role of HuR in demyelinating diseases. The novel therapeutic approach using anti-HuR ASO was introduced and explained. The topic is interesting and important in the field.
However, this reviewer found that the manuscript is not well organized and reader unfriendly in the following sense:
(1) The length of each paragraph is too long, especially, sections 2, 3, 4, and 5. (2) There is a table (Table 1.) but not explained well and quoted effectively in the main text. (3) There are no figures to help the readers to understand the topic and follow the story.
These issues should be addressed.
Author Response
The authors reviewed the physiological role of HuR outside and inside the nervous system, and the relationship of HuR with neurodegenerative disorders. They have especially focused on the role of HuR in demyelinating diseases. The novel therapeutic approach using anti-HuR ASO was introduced and explained. The topic is interesting and important in the field.
We thank the reviewer for his/her kind comments.
However, this reviewer found that the manuscript is not well organized and reader unfriendly in the following sense:
(1) The length of each paragraph is too long, especially, sections 2, 3, 4, and 5.
We thank the reviewer for bringing this to our attention and we divided sections in more paragraphs to facilitate the fluency of the article.
(2) There is a table (Table 1.) but not explained well and quoted effectively in the main text.
We increased the citation and explanation of Table 1 in the main text.
(3) There are no figures to help the readers to understand the topic and follow the story. These issues should be addressed.
We thank the reviewer for the suggestion, we added a Graphical Abstract in the text.

Reviewer 3 Report
This is a timely and nice review about the RNA-Binding protein HuR on neurodegenerative diseases. The structure and content are all well organized and well written. References well cited. I have only a couple suggestions for the authors.
(1). The title ‘Targeting RNA binding protein as a novel therapeutic approach…” suggests a large part of the review would be devoted on the “therapeutic approaches”. However, there are not much such approaches presented throughout. The authors may consider add a section or change the title to make it consistent.
(2). Section 3 is a huge paragraph. It would be clearer to read if this section could be divided into different paragraphs dealing with microglia, astrocytes, the role of HuR.
(3). For section 4, it also would be good to summarize the commonalities and specificities of HuR in different degenerative disorders at the end of the section.
Author Response
Reviewer n. 3
This is a timely and nice review about the RNA-Binding protein HuR on neurodegenerative diseases. The structure and content are all well organized and well written. References well cited. I have only a couple suggestions for the authors.
(1). The title ‘Targeting RNA binding protein as a novel therapeutic approach…” suggests a large part of the review would be devoted on the “therapeutic approaches”. However, there are not much such approaches presented throughout. The authors may consider add a section or change the title to make it consistent.
We thank the reviewer for the suggestion, we changed the title accordingly.
(2). Section 3 is a huge paragraph. It would be clearer to read if this section could be divided into different paragraphs dealing with microglia, astrocytes, the role of HuR.
Following the suggestion of the reviewer, we divided the paragraph in two sections.
(3). For section 4, it also would be good to summarize the commonalities and specificities of HuR in different degenerative disorders at the end of the section.
We thank the review for the suggestion, we added a summary of the effect in the conclusion section.

Round 2
Reviewer 2 Report
The authors have responded to the reviewers comments accordingly.